# A Methodological Quality Assessment of Meta-Analysis Studies in Dance Therapy Using AMSTAR and AMSTAR 2

**DOI:** 10.3390/healthcare8040446

**Published:** 2020-11-01

**Authors:** Hye-Ryeon Kim, Chang-Hwan Choi, Eunhye Jo

**Affiliations:** 1Graduate School of Education, Major of Dance Education, Dongguk University, Seoul 04620, Korea; hyeryeon0902@dongguk.edu; 2Department of Sports, Health, and Rehabilitation, Kookmin University, Seoul 02707, Korea; 3ICT Convergence Research Center, Soonchunhyang University, Asan 31538, Korea

**Keywords:** dance therapy, meta-analysis, AMSTAR, AMSTAR 2, methodological, quality assessment

## Abstract

Although earlier meta-analysis studies have provided evidence-based information useful for decision-making, debate regarding their quality continues. This study aimed to evaluate the quality of meta-analysis studies in the field of dance therapy (DT) using the Assessment of Multiple Systematic Reviews (AMSTAR) and AMSTAR 2 assessment tools. Meta-analysis studies on DT were collected from various databases. Seven meta-analysis studies were selected for this study. Our findings showed that the quality level of the meta-analysis studies related to DT was “High” on the AMSTAR evaluation, but their quality decreased to “Low” on the AMSTAR 2 evaluation. Moreover, using AMSTAR 2, 71.43% of the studies fell within the category of “Moderate” or below. There was no statistically significant difference in the quality scores of the characteristics of these studies. Our results suggest that (1) education on meta-analysis guidelines is required to improve the quality of DT-related meta-analysis studies, and (2) methodological caution is warranted, since different outcomes in evaluation scores for each tool may be obtained when using AMSTAR and AMSTAR 2. Based on this study, it is expected that common and specific guidelines for meta-analysis in DT can be established.

## 1. Introduction

Dance is one of the oldest therapy methods [1]. Dance therapy (DT) is a treatment method to assist the physical, emotional, and cognitive recovery of an individual or a group by using dance movements [2]. DT has many differences compared to general dance. DT is more than just “exercise”, in fact, it allows the patient to communicate complex and difficult to verbalize emotions through the performance of dance movements, which are then carefully interpreted and assessed by the DT specialist. It is a multi-layered and complex treatment program in which the therapist composes different body movements and determines the tempo of music, the therapist then observes and diagnoses the patient’s emotions and physical condition [3].

DT had been applied as an alternative therapy for simple physical rehabilitation and functional recovery in the past, but it has recently been reported as an effective treatment method for stress and depression, and has also been effective as exercise, for recovery, health and fitness, and social development [4]. In particular, DT is used to treat individuals who cannot freely control their movements, such as individuals with Parkinson’s disease and development disorders. DT is highly effective for functional development and psychological stability. Moreover, it is used as an official treatment program for Parkinson’s disease in 25 countries. Additionally, it provides entertainment value through dancing and a combination of various body movements [5,6]. The number of DT participants is increasing continually, as DT is a broad-spectrum treatment accessible for people of various ages, from children to the elderly [7,8]. University programs to educate instructors are also becoming more common [9,10].

The effects and advantages of DT have been reported in various fields, such as medicine, nursing, physical education, dance, and pedagogy [1,5,11,12,13,14,15]. As many studies on the effectiveness and advantages of DT have been published, an evidence-based practice (EBP) has been undertaken in academia to test the actual effectiveness of DT. EBP is a scientific approach to assess decision-making by establishing theoretical evidence based on research results and the interpretation of phenomena according to systematic norms [16]. It includes various scientific approaches, such as clinical observation with scientific validity, qualitative research, systematic case study research, single-subject experiments, wireless allocation control, and meta-analysis research [17].

Meta-analysis is a quantitative research method that estimates average or common effects by statistically combining two or more individual studies on a specific research topic [18]. Moreover, the results of research integration through meta-analysis form a very strong scientific basis [19,20]. In particular, meta-analysis is very useful for clinical decisions in health and medical fields because it enables quantitative measurement and comparison of the effects of treatment and uncertainty [21]. For meta-analysis to be utilized at a high level of evidence, the reliability and validity of research methods and results, as well as quantitative increase, must be guaranteed first [22]. Nevertheless, it is reported that meta-analysis has the potential to generate inaccurate conclusions because it synthesizes individual studies with inconsistently high or low homogeneity and validity. As reasons for this criticism, authors often cite the risk for illogical comparison between different samples and measurement methods for meta-analysis, the problem of using low-quality data, the problem of deviations in research selection, and the problem of violation of the assumption of independence [23,24]. Therefore, before admitting the meta-analysis results, it is essential to test whether or not data collection and statistical analysis are conducted properly, from a methodological standpoint. Accordingly, an evaluation of the quality of meta-analysis research has been proposed as a strategy to overcome the common pitfalls of meta-analysis and to ensure reliable results [25,26,27].

Various assessment tools have been developed and used to evaluate the quality of meta-analysis research. Since the early 1990s, the Potsdam International Consultation on Meta-analysis (PCM) [28], Quality of Reporting of Meta-analyses (QUOROM) [24], Meta-analysis of Observation Studies in Epidemiology (MOOSE) [29], Preferred Reporting Items for Systematic Reviews and Meta-analysis (PRISMA) [28], the Overview Quality Assessment Questionnaire (OQAQ) [30], and the Assessment of Multiple Systematic Reviews (AMSTAR) [17,31] assessment tools have been used to conduct quality assessment of meta-analysis studies [26]. The PCM, developed in the early 1990s, has comparatively fewer items and focuses only on particular scientific research methodologies [17]. Relatively speaking, QUOROM assesses the quality of research systemically but compared with the PCM, it has the disadvantage of comprising several evaluation items and it takes a long time to perform [17]. To make up for this weakness, PRISMA—comprising 27–35 items that evaluate the quality of observational research—was developed; however, its application is limited depending on the research design of the study, and it is currently used only as a reporting standard for systematic literature reviews [17,31].

Conversely, AMSTAR is the only assessment tool with proven reliability and validity that has been developed and revised recently through the supplementation of evaluation contents and methods based on the existing OQAQ and Sack’s checklist [17,31]. The AMSTAR assessment tool consists of eleven items, including the establishment of a literature review plan, literature selection, and data extraction method, as well as a comprehensive literature search, inclusion criteria for publication status, presentation of included research lists, presentation of research characteristics, evaluation of research quality, use of research quality to draw conclusions, appropriateness of the combining method of individual research results, evaluation of the publication possibility, and description of conflicts of interest. Compared to existing tools, AMSTAR can systematically assess the quality of meta-analyses while testing their research methodology [17]. Moreover, AMSTAR has the advantage of being able to objectively compare and analyze the level of quality of meta-analysis studies because its evaluation items are concise, easy to apply, and can be scored on a binary 0 or 1 scale. AMSTAR is widely used around the world in various academic disciplines, including health and medical fields [17,25,26,27,31].

Based on the advantages of AMSTAR, AMSTAR 2 was recently modified and released. Both are tools for evaluating methodological quality, but they differ in terms of the number of evaluation items, evaluation methods, and evaluation of results and calculation methods. Specifically, AMSTAR 2 consists of 16 evaluation items—five more items than AMSTAR—and its evaluation method includes not only the existing binary system (“Yes” and “No”), but also a midway point (“Partial Yes”), enabling a more detailed assessment. Moreover, there is a huge difference in calculating evaluation results between the two. For example, unlike AMSTAR that calculates the combined scores of eleven items, AMSTAR 2 calculates the scores of the evaluation by considering the number of important questions omitted.

Therefore, it is not clear whether similar methodological quality evaluations can be performed for the same research because AMSTAR and AMSTAR 2 have a different amount of evaluation items, different item contents, evaluation methods, and evaluation results calculation methods [32]. Furthermore, in the guidelines for conducting an overview of the Cochrane Collaboration’s Systematic Reviews (SR), AMSTAR is mentioned as the only available tool. Thus, in order to use AMSTAR 2 as a tool for SR, research is needed to confirm its use [33]. Using the updated version of the assessment tool is very important in that it can achieve a clearer evaluation by applying the improvement made from the previous assessment tool to new research and confirm its validity.

For sustainable research in the field of DT, a SR of the quality of meta-analysis studies is required. Although research has been conducted to evaluate the quality of systematic literature reviews in the field of dance, to our knowledge, no studies have been conducted on the methodological quality evaluation of meta-analysis related to DT. Hence, in this study, AMSTAR and AMSTAR 2 are used simultaneously to perform a methodological quality evaluation of meta-analysis studies and analyze the assessment results generated by each tool. Therefore, this study will test the following: (1) What are the main characteristics of DT meta-analysis? (2) What is the methodological quality evaluation score and quality level of DT related meta-analysis research? and (3) Is there a difference in the quality of evaluation score for each variable of DT in the meta-analysis research?

Overall, this study attempts to analyze the quality level and score of each study by identifying the main characteristics of DT-related meta-analysis studies using the AMSTAR and AMSTAR 2 evaluation tools, and evaluating the methodological quality. Based on this, it is possible to check the difference of the scores according to the variables of each study, and by comparing the scores according to the evaluation tools, basic data for the improvement of the quality of subsequent studies will be provided.

## 2. Methods

### 2.1. Data Sources

For data collection, meta-analysis studies on DT were selected as the statistical population. Web of Science, PubMed, Google Scholar, and Elsevier were selected as the academic databases (DBs) for data collection. The inclusions were as follows: (1) Meta-analysis studies related to dance therapy, (2) studies in which information of the size of the effect was presented as a quantitative study; (3) studies that were presented as research papers written in English, and (4) open studies that were available online. In addition, in the case of overlap between degree-oriented research (such as dissertations) and academic journals, the latter were used as a basis. Based on these selection criteria, three research directors (one of whom majored in Dance and the other two in Measurement and Evaluation) each prepared a list of selected studies through data collection. The research directors conducted a systematic search between May and July 2020 and selected research studies from the aforementioned DBs until 30 August 2020.

Moreover, a list was prepared comprising studies that included the following words in their titles and keywords (main themes): [Dance Therapy, DMT, Meta, Meta-Analysis], and [Dance Therapy Meta, Dance Therapy Meta-Analysis, DMT Meta]. Precaution was taken to ensure that there were no missing research materials when selecting the studies. In addition to the search of research materials, a secondary search was performed, following the citations of the selected research materials. The final subjects of the studies were selected after consensus had been reached based on discussion of the selection criteria and the list of research materials prepared by the three researchers. A total of 44 individual studies on meta-analysis related to dance therapy were collected, and seven (from academic journals) were selected as the final individual studies, excluding 32 overlapping studies and five that were not available online or that did not provide original texts. The specific details are shown in Figure 1 below.

### 2.2. Assessment Tool

For methodological quality evaluation, the AMSTAR and AMSTAR 2 assessment tools were used. First, AMSTAR was developed by Shea in 2007 [34], and it consists of 11 items that evaluate the appropriateness of methodology. Items are evaluated as “Yes” (meaning the item has been properly handled, 1 point), “No” (indicating the possibility that the item did not perform well, 0 points), and “Not applicable” (in case of performance failure because the item was not applied, 0 points). AMSTAR is also an open tool that can be used without special consent and is included in the manual for systematic consideration of the NECA (National Evidence-Based Healthcare Collaborating Agency). In particular, AMSTAR 2 provides evidence for validity and reliability as a methodological quality assessment tool of meta-analysis research [35]. AMSTAR is widely used as a valuable tool to evaluate the quality of a meta-analysis study conducted in any academic field. Details of the AMSTAR assessment tool is shown in Table 1.

In 2017, AMSTAR 2, a revised version of AMSTAR, was announced [36]. It consists of 16 items, graded as “Yes,” “Partial Yes,” and “No” instead of “Yes” or “No.” Specifically, “Yes” is selected when the reporting and implementation of each item are satisfactory. “Partial Yes” is selected when the reporting and implementation of each item are insufficient. Finally, “No” is selected when there is no reporting and implementation of the item. The AMSTAR 2 assessment tool is something that has not been employed in many academic disciplines yet. In the field dance-related studies, AMSTAR 2 has been rarely used. Details of the AMSTAR 2 assessment tool are shown in Table 2.

In the case of AMSTAR 2, depending on the number of important items missing out of all 16 questions (items 2, 4, 7, 9, 11, 13, 15), the assessment is as follows: “Critically low quality,” “Low quality,” “Moderate quality,” and “High quality” [36]. Thus, by using AMSTAR 2, evaluation results are calculated based on the number of important and non-essential items omitted, rather than the total score of evaluation items. Moreover, AMSTAR 2 has the advantage of being able to derive more specific and clear results because “Partial Yes” is included in the selection, rather than the binary construct of “Yes” and “No.”

### 2.3. Procedure

First, the two researchers with experience in meta-analysis studies and awareness of all items in AMSTAR and AMSTAR 2 individually assessed the studies to understand the content of the items of AMSTAR and AMSTAR 2 and to perform the data coding accurately. Second, after the researchers carried out individual evaluations and compared the assessment outcomes, the original evaluation data were used to assess the results of the same evaluation [37]. In addition, through implementing Cohen’s kappa coefficient analysis—which checks the degree of consistency between measurement category values—the consistency of the two researchers’ findings was confirmed. In general, the Kappa coefficient is interpreted as “Slight” (slight agreement) when it has a value between 0.0 and 0.2, “Fair” (a certain degree of consistency) for a value between 0.2 and 0.4, “Moderate” (appropriate agreement) for a value between 0.4 and 0.6, “Substantial” (high agreement) for a value between 0.6 and 0.8, and “Almost Perfect” (nearly perfect agreement) for a value between 0.8 and 1.0 [38]. This study tested the consistency between the AMSTAR and AMSTAR 2 assessment tools. Cohen’s Kappa for AMSTAR’s consistency was 0.882 and 0.903 for AMSTAR 2, indicating that both assessment tools showed an almost perfect agreement. We believe that this outcome was possible because of the prior discussion of research experiences and the reach of consensus between the two researchers, as well as because of their understanding of each evaluation item and range of measurements.

### 2.4. Data Analysis

The analysis methods used to solve the research problem set forth by this study were the following. Descriptive statistics was performed to determine the characteristics of the final data and to check their quality evaluation score. Moreover, in order to confirm the assessment consistency between the researchers, the Mann-Whitney U test, a non-parametric statistical method, was conducted to analyze Cohen’s kappa coefficients and to compare quality evaluation scores of all research characteristic variables (year of publication, number of research materials, research fund support, research period, and research field). The Mann-Whitney U test was performed because the basic assumptions of parametric statistics were not satisfied. For data analysis, IBM SPSS v25.0 software was used, and the statistical significance level was set to 0.05. The overall procedure for this study is shown in Figure 2 below.

## 3. Results

### 3.1. The Main Characteristics of Meta-Analysis Research Related to Dance Therapy

The final study data included seven papers, as shown in Table 3. Our research revealed that a meta-analysis of studies related to dance movement therapy for children and adult psychiatric patients began in 1996, and after that, two journals in 2014, one in 2016, and three in 2019 were published. The number of individual studies included in the meta-analysis ranged from 2 to 41. Regarding the duration of these meta-analysis studies, five were conducted for less than 10 years, whereas two of them were conducted for more than 10 years. Moreover, the average number of authors of the studies ranged from 2 to 6. Two studies received funding and five did not. By study subject, we studied three meta-analysis studies to confirm the effectiveness of dance therapy in patients with chronic heart failure, Parkinson’s disease, high blood pressure, and mental disorders (e.g., physical disorders, mental retardation). Two meta-analysis studies confirmed the effect of DT on mental health (including depression) and quality of life. Moreover, two studies tested the effectiveness of DT through a randomized controlled experiment without any intervening variables. Overall, we found that dance movement therapy was used to treat both physical and mental illnesses. In particular, it was used to treat patients with Parkinson’s disease and heart and blood pressure issues, as well as to improve the quality of life, mood, and to reduce depression.

### 3.2. Methodological Quality Assessment Score and Level of Meta-Analysis Related to Dance Therapy

Table 4 and Table 5 show the comparison of methodological quality evaluation of meta-analysis studies related to dance therapy by each assessment tool. The AMSTAR score for a total of seven individual studies ranged from 4 to 10, with one study (14.29%) rated as “Low” and six studies (85.71%) rated as “High.” The overall result of the evaluation was deemed to be “High.” The average score was 9.0. The AMSTAR 2 score ranged from 6.5 points to 14 points, and the average score was 12.2. There were three studies rated as “Low” (42.86%), two studies rated as “Critically Low” (28.57%), and two studies rated as “Moderate” (28.57%). The overall rating appeared to be “Low.”

The contents of the methodological quality evaluation results of meta-analysis studies related to DT by each tool are shown in Figure 3.

Table 6 includes the content of important questions in AMSTAR 2 and the probability of including these items in the final data collected in this study. In the case of item 7, the probability was at 28.5%, indicating the lowest inclusion probability in the final study of meta-analysis studies related to DT. The next lowest inclusion probability was 71.4% for item 13, followed by 78.5% for item 15 and 85.7% for item 9. In the case of questions 2, 4, and 11, the probability was 100%.

### 3.3. Testing the Differences in Quality Evaluation Scores in Meta-Analysis Studies Related to Dance Therapy

Table 7 shows the difference between the group means of AMSTAR’s quality evaluation scores according to the characteristics of meta-analysis studies related to DT. Table 8 shows the difference between the group means of quality evaluation scores of AMSTAR 2. The classification criteria for the research characteristic variables selected for this study were as follows. The standard for publication year was 2016, and the standard number of studies included was 15. Research funding was classified as “yes” or “no,” and the duration of the research period was classified as “1–6 years” or “more than 6 years.” Lastly, the research field was classified based on whether the “exercise” variable was the only match for the keywords selected for our study, or whether other variables, such as psychological state and mood, were presented in addition to “exercise.” As shown in Table 7, there were no statistically significant differences in publication year, number of research articles included, funding, research period, and research field. Similarly, Table 8 shows that there were no statistically significant differences in any of the five meta-analysis research characteristic variables.

## 4. Discussion

This study was conducted to propose a way to improve the methodological quality of SR-based meta-analysis through a qualitative evaluation of meta-analysis studies in the field of DT. In particular, the purpose of this study was to provide a variety of information from a methodological perspective by applying assessment tools with proven reliability and validity (such as AMSTAR and AMSTAR 2). Seven studies were selected, according to the inclusion criteria. Meta-analysis studies in the field of DT began to be published in 1996, and in 2019, three papers were published, which was the highest number of related studies to come out in a single year. This indicates a growing academic interest in DT as more SR-based meta-analysis studies are being published [1,12,13]. Through our study, the quality of meta-analysis studies assessed using AMSTAR was found to be “High”, with an average of 9.0 points out of 11. Conversely, using AMSTAR 2, the quality of the aforementioned studies was found to be “Low”, with an average of 12.2 points out of 16. These results are similar to the findings of meta-analysis quality assessment of healthcare interventions using AMSTAR (9.6) and were higher than that of the quality evaluation in healthcare using AMSTAR and AMSTAR 2 (mean = 6.9, and mean = 7.95, respectively) [32].

The overall evaluation results of this study showed that AMSTAR 2 produced a lower rating than AMSTAR. This can be explained by the omission of items 7, 9, 13, and 15 in AMSTAR 2. Using AMSTAR 2, the overall quality assessment becomes very low when some of the important questions are omitted [32]. In particular, the item with the highest omission rate was item 7, which was included in only two of the total seven studies. The items with the next highest omission rates were items 13 (included only in five studies), 9 and 15 (which were included in six of seven studies).

There are many opinions on the overall quality evaluation and interpretation of AMSTAR and AMSTAR 2, especially regarding their difference in methodological quality scores [32]. We found that studies that received high ratings on AMSTAR received lower ratings on AMSTAR 2. One evaluation varied significantly, from a “High” score to “Critically Low”, signaling the wide gap in results. This is because the quality assessment can become very low if any of the important items in AMSTAR 2 are omitted.

When we used AMSTAR 2 to calculate the probability of the data in the study containing important questions, item 7 showed the lowest inclusion probability at 28.5%. This item evaluates whether or not a list of individual studies excluded from the research method and results section was presented. It specifically assesses whether or not the list of studies excluded from meta-analysis was presented in the form of a table, and evaluates all studies briefly describing excluded studies as “No”. According to the assessment in this study, only two of seven studies were found to have item 7. Therefore, the list of excluded studies and the reason for the exclusion must be explained since if the list of excluded studies is not provided and the reasons for the exclusion are not explained at the same time, the influence of the excluded studies on a systematic literature review cannot be determined [36].

The item that had the second-lowest probability of inclusion, 71.4%, was item 13. The item evaluates whether or not the risk of bias (RoB) of individual studies was explained in the results and discussion sections. For example, if the individual studies selected in the meta-analysis included randomized control trials (RCT) or non-randomized control trials (NRCT), item 13 assesses whether the effect of RoB on the outcome was described. According to the assessment, five out of seven were found to include item 13. Next, item 15 and item 9 showed 78.5% and 85.7% of inclusion probability, respectively. Item 15 and 9 are both evaluation items on publication bias and RoB, focusing on the objectivity of data gathering and result reporting. In some cases, they might not be present, depending on the amount of research data excluded from individual study products of meta-analysis or at the discretion of researchers. However, meta-analysis is considered the most powerful type of evidence in evidence-based medicine (EBM) and has a large influence on health-related policy-making and decision-making in clinical settings [39]. Consequently, it is very important to increase the quality of meta-analysis to minimize the possibility for inaccurate assumptions and conclusions [40].

There was no statistically significant difference between groups in both AMSTAR and AMSTAR 2 after testing the difference in quality evaluation according to the research characteristics. These results confirm that DT-related meta-analysis studies do not exhibit differences due to research characteristics regardless of whether AMSTAR or AMSTAR 2 is used. In previous studies, the “recent study” category showed a higher average value if the difference was verified based on the “year” variable [34,35]. Moreover, recent studies have reported that PRISMA or AMSTAR standards perform well. However, our results suggest that the quality of meta-analyses has decreased recently. Thus, various strict review and training standards should be applied during the publication process to improve the quality of meta-analysis studies.

To perform a meta-analysis, a quality evaluation of the subject paper must be carried out prior, as it is closely connected to the reliability of the research results [22]. The reliability and validity of meta-analyses increase when they evaluate the quality of the paper taking into account the research design, quality of the literature as well as the quantity, consistency, and directness of evidence, and whether the study incorporates these factors to present a comprehensive conclusion [23]. However, using AMSTAR 2, 71.43% of the studies were identified as less than “Moderate,” which limited the interpretation of the meaning of a meta-analysis due to insufficient quality of the target paper. Since it is becoming more common to evaluate papers using quality assessment tools, it is believed that these tools will contribute to improving the quality of meta-analysis research if the quality assessment criteria are officially introduced in the journal’s review criteria.

In many instances, quality assessment of individual studies is insufficient and, in some cases, this is because quality assessment tools were not properly used [22]. In this respect, this study is academically significant because it is a first attempt to evaluate the quality of meta-analysis studies related to DT. In particular, AMSTAR and AMSTAR 2 were used concurrently to conduct a methodological quality assessment of meta-analyses in the DT field. Based on these results, our study intends to provide information for improving the quality of meta-analysis studies. However, since this study did not consider various qualitative levels of individual studies and only used AMSTAR and AMSTAR 2, there is a limit to the interpretation. If various qualitative assessment tools are applied in future research, a more reasonable qualitative evaluation of meta-analyses in the DT field can be performed.

SR in the DT field is becoming more important because it provides a significant amount of information for decision-making. The methodological deficiencies of SRs are linked to the research results, causing a significant decrease in validity. Therefore, the development of appropriate critical analysis and evaluation tools to conduct SRs is paramount. In subsequent meta-analysis studies in various fields, a more detailed and valid evaluation will be possible if the methodological quality evaluation results are described using the newly developed AMSTAR tool.

## 5. Conclusions

This study aimed to evaluate the quality level of meta-analysis studies related to DT using AMSTAR and AMSTAR 2 to suggest methodological quality improvement measures. Regarding DT-related meta-analysis studies, their qualitative level was found to be “High” based on the AMSTAR evaluation standard, and “Low” based on the AMSTAR 2 standard. In the case of AMSTAR 2, 71.43% of studies were included in the category as ‘Moderate’ or below. No difference was found at any statistically significant level when the qualitative scores according to the study characteristics were compared. Through this study, we obtained the following findings: (1) Utilization of public education on meta-analysis guidelines is required to improve the quality of meta-analysis studies related to DT, while strict standards are required in the publication review process of meta-analysis studies; (2) when using AMSTAR and AMSTAR 2, caution in their application is required because there may be differences in evaluation scores depending on the tool used. Based on this study, we look forward to preparing and sharing common and specific guidelines for meta-analysis studies related to DT.

## Figures and Tables

**Figure 1 healthcare-08-00446-f001:**
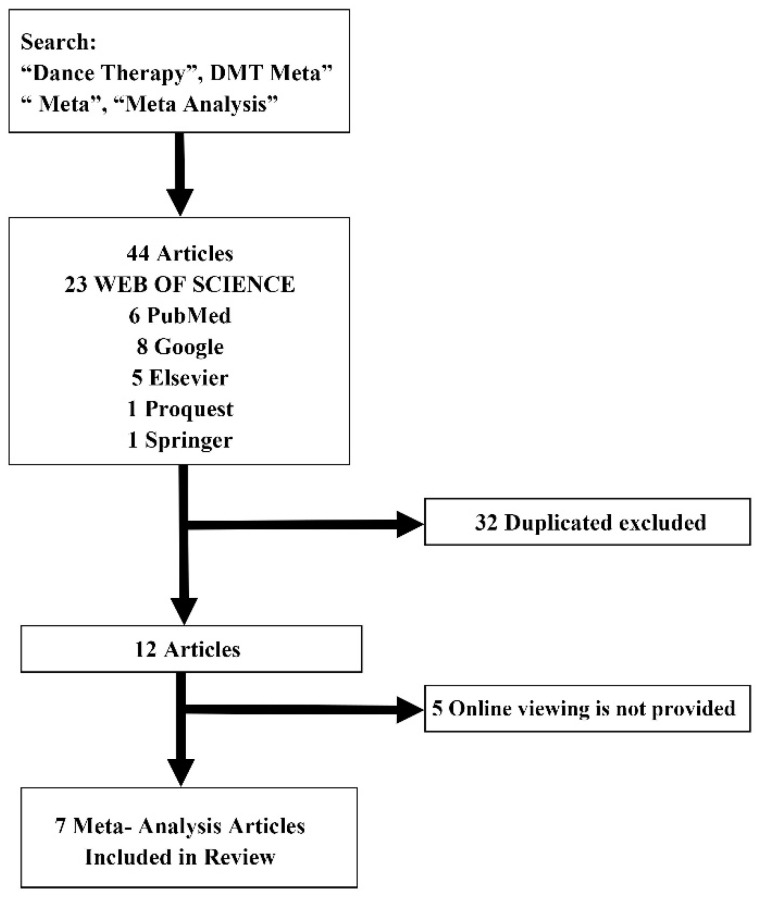
Meta-analysis related to dance therapy selection process of individual research products.

**Figure 2 healthcare-08-00446-f002:**
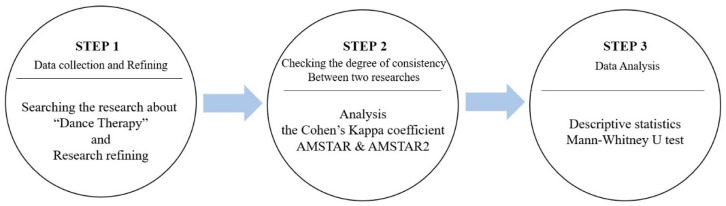
The overall procedure for this study.

**Figure 3 healthcare-08-00446-f003:**
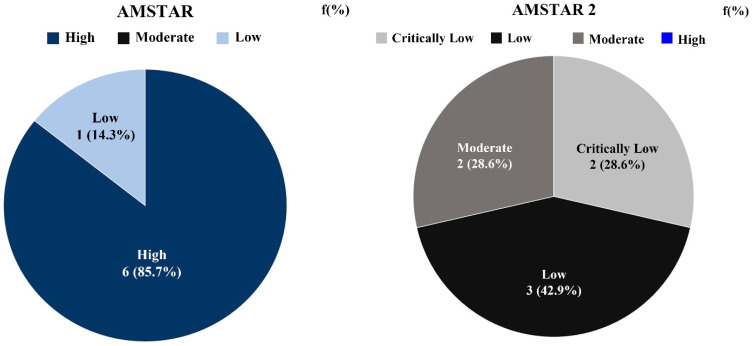
Quality evaluation according to the assessment tool of meta-analysis research related to dance therapy.

**Table 1 healthcare-08-00446-t001:** Question content of the assessment of multiple systematic reviews (AMSTAR).

Question Content
1. Was an ‘a priori’ design provided?
2. Was there a duplicate study selection and data extraction?
3. Was a comprehensive literature search performed?
4. Was the status of publication (i.e., grey literature) used as an inclusion criterion?
5. Was a list of studies (included and excluded) provided?
6. Were the characteristics of the included studies provided?
7. Was the scientific quality of the included studies assessed and documented?
8. Was the scientific quality of the included studies used appropriately in formulating conclusions?
9. Were the methods used to combine the findings of studies appropriate?
10. Was the likelihood of publication bias assessed?
11. Was the conflict of interest stated?

Evaluation (Yes: 1 point, No: 0 point, Not applicable: 0 point). Aggregated Score Analysis (0–4 points: Low level; 5–8 points: Moderate level; 9–11 points: High level).

**Table 2 healthcare-08-00446-t002:** Question content of the AMSTAR 2.

Question Content
1. Did the research questions and inclusion criteria for the review include the components of PICO?
2. Did the report of the review contain an explicit statement that the review methods were established prior to the conduct of the review and did the report justify any significant deviations from the protocol?
3. Did the review authors explain their selection of the study designs for inclusion in the review?
4. Did the review authors use a comprehensive literature search strategy?
5. Did the review authors perform a study selection?
6. Did the review authors perform data extraction in duplicate?
7. Did the review authors provide a list of excluded studies and justify the exclusions?
8. Did the review authors describe the included studies in adequate detail?
9. Did the review authors use a satisfactory technique for assessing the risk of bias (RoB) in individual studies that were included in the review?
10. Did the review authors report on the sources of funding for the studies included in the review?
11. If meta-analysis was performed, did the review authors use appropriate methods for statistical combination of results?
12. If meta-analysis was performed, did the review authors assess the potential impact of RoB in individual studies on the results of the meta-analysis or other evidence synthesis?
13. Did the review authors account for RoB in primary studies when interpreting/discussing the results of the review?
14. Did the review authors provide a satisfactory explanation for, and discussion of, any heterogeneity observed in the results of the review?
15. If they performed quantitative synthesis did the review authors carry out an adequate investigation of publication bias (small study bias) and discuss its likely impact on the results of the review?
16. Did the review authors report any potential sources of conflict of interest, including any funding for conducting the review?

Evaluation (Yes: 1 point; PY: 0.5 point; No: 0 point). Aggregated Score Analysis: Critically low quality, low quality, moderate quality, high quality important items 2, 4, 7, 9, 11, Questions 13 and 15 are of very low quality if two or more of these items are omitted, low quality if one important item is omitted, moderate quality if two or more of the non-critical items other than the above important items are omitted, no missing items or non-important. If one item is omitted, it is rated as excellent quality.

**Table 3 healthcare-08-00446-t003:** Main features of meta-analysis research related to dance therapy studies.

No.	Publication Year	Number of Authors	Topic Area (Keyword)	Period of Meta-Analysis	Number of Studies Analyzed	Research Fund Support
1	2019	6	Dance Movement Therapy, Dance interventions, Meta-Analysis, Randomized controlled trial, Clinical controlled trial, Creative art therapies, Integrative medicine	2012–2018	41	Yes
2	2019	5	Blood Pressure, Dance, Exercise, Meta-Analysis	2013–2018	7	No
3	2019	4	Dance Movement Therapy, Depression, Effectiveness, Systematic review, Meta-Analysis	2005–2018	8	Yes
4	2016	5	Dance, Parkinson’s disease, PD, Cognition, Mood, Meta-Analysis	2012–2015	4	No
5	2014	3	Exercise tolerance, Quality of life, Cardiac failure, Dance	2008–2013	2	No
6	2014	4	Dance Movement Therapy, Therapeutic use of dance, Meta-Analysis, review of evidence-based research, Randomized controlled trials, Integrative medicine	2006–2012	23	No
7	1996	2	Children, Adult Psychiatric Patients, Mentally Retarded/Physically Disabled, Elderly	1974–1993	23	No

**Table 4 healthcare-08-00446-t004:** Quality evaluation using AMSTAR.

Study No.	AMSTAR Evaluation Questions											Total	Overall Rating
	Q1	Q2	Q3	Q4	Q5	Q6	Q7	Q8	Q9	Q10	Q11		
1	Yes	Yes	Yes	Yes	No	Yes	Yes	Yes	Yes	Yes	No	9 (82%)	high
2	No	Yes	Yes	Yes	Yes	Yes	Yes	Yes	Yes	Yes	No	10 (91%)	high
3	Yes	No	Yes	Yes	Yes	Yes	Yes	Yes	Yes	Yes	Yes	10 (91%)	high
4	Yes	Yes	Yes	Yes	No	Yes	Yes	Yes	Yes	Yes	Yes	10 (91%)	high
5	Yes	Yes	Yes	Yes	No	Yes	Yes	Yes	Yes	Yes	Yes	10 (91%)	high
6	Yes	Yes	Yes	Yes	No	Yes	Yes	Yes	Yes	Yes	Yes	10 (91%)	high
7	Yes	No	Yes	No	No	Yes	No	No	Yes	No	No	4 (36%)	Low
Mean (SD)	1.0 (0.0)	0.7 (0.5)	1.0 (0.0)	0.9 (0.4)	0.3 (0.5)	1.0 (0.0)	0.9 (0.4)	0.9 (0.4)	1.0 (0.0)	0.9 (0.4)	0.6 (0.5)	9.0 (2.2)	high

**Table 5 healthcare-08-00446-t005:** Quality evaluation using AMSTAR 2.

Study No.	AMSTAR 2 Evaluation Questions																Total	Overall Rating	Critical Weaknesses Item
	Q1	Q2	Q3	Q4	Q5	Q6	Q7	Q8	Q9	Q10	Q11	Q12	Q13	Q14	Q15	Q16			
1	Yes	Yes	Yes	Yes	Yes	Yes	No	Yes	Yes	No	Yes	Yes	Yes	Partial Yes	Yes	No	12.5 (78%)	Low	7
2	Yes	Yes	Yes	Yes	Yes	Yes	Yes	Yes	Yes	No	Yes	Yes	Yes	Yes	Yes	No	14 (88%)	Moderate	-
3	Yes	Yes	Yes	Yes	No	Yes	Yes	Yes	Yes	No	Yes	Yes	Yes	Yes	Yes	Yes	14 (88%)	Moderate	-
4	Yes	Yes	Yes	Yes	Yes	Yes	No	Yes	Yes	No	Yes	Yes	No	Partial Yes	No	Yes	11.5 (72%)	Critically Low	7, 13, 15
5	Yes	Yes	Yes	Yes	Yes	Yes	No	Yes	Yes	No	Yes	Yes	Yes	Partial Yes	Yes	Yes	13.5 (84%)	Low	7
6	Yes	Yes	Yes	Yes	Yes	Yes	No	Yes	Yes	No	Yes	Yes	Yes	Partial Yes	Yes	Yes	13.5 (84%)	Low	7
7	Yes	Yes	Yes	Yes	No	No	No	Yes	No	No	Yes	No	No	No	Partial Yes	No	6.5 (41%)	Critically Low	7, 9, 13
Mean (SD)	1.0 (0.0)	1.0 (0.0)	1.0 (0.0)	1.0 (0.0)	0.7 (0.5)	0.9 (0.4)	0.3 (0.5)	1.0 (0.0)	0.9 (0.4)	0.0 (0.0)	1.0 (0.0)	0.9 (0.4)	0.7 (0.5)	0.6 (0.3)	0.8 (0.4)	0.6 (0.5)	12.2 (2.7)	Low	

**Table 6 healthcare-08-00446-t006:** Probability of including AMSTAR 2 important items in individual studies.

Item No.	Important Question Content	Probability of Response
Item 2	Did the report of the review contain an explicit statement that the review methods were established prior to the conduct of the review and did the report justify any significant deviations from the protocol?	100%
Item 4	Did the review authors use a comprehensive literature search strategy?	100%
Item 7	Did the review authors provide a list of excluded studies and justify the exclusions?	28.5%
Item 9	Did the review authors use a satisfactory technique for assessing the risk of bias (RoB) in individual studies that were included in the review?	85.7%
Item 11	If meta-analysis was performed, did the review authors use appropriate methods for a statistical combination of results?	100%
Item 13	Did the review authors account for RoB in primary studies when interpreting/discussing the results of the review?	71.4%
Item 15	If they performed quantitative synthesis, did the review authors carry out an adequate investigation of publication bias (small study bias) and discuss its likely impact on the results of the review?	78.5%

**Table 7 healthcare-08-00446-t007:** Differences in the AMSTAR quality score according to research characteristics.

Characteristics	Type	N (%)	M ± SD	Mann-Whitney (*sig*)
Year	1996–2016	4 (57.1)	9.7 ± 0.6	6.00 (1.00)
	2017–2019	3 (42.9)	8.5 ± 3.0	
Studies	15 less	4 (57.1)	10.0 ± 0.0	2.0 (0.078)
	15 more	3 (42.9)	7 ± 3.2	
Fund support	yes	2 (28.6)	9.5 ± 0.7	4.0 (0.629)
	no	5 (71.4)	8.8 ± 2.7	
Research period	1–6 years	3 (42.9)	10.0 ± 0.0	5.0 (0.659)
	More than 6 years	4 (57.1)	8.3 ± 2.9	
Research field	Only exercise related	3 (42.9)	8.0 ± 3.5	3.0 (0.186)
	Including exercise	4 (57.1)	9.8 ± 0.5	

**Table 8 healthcare-08-00446-t008:** Differences in the AMSTAR 2 quality score according to research characteristics.

Characteristics	Type	N (%)	M ± SD	Mann-Whitney (*sig*)
Year	1996–2016	4 (57.1)	11.3 ± 3.3	2.0 (0.150)
	2017–2019	3 (42.9)	13.5 ± 0.9	
Studies	15 less	4 (57.1)	13.3 ± 1.2	2.5 (0.208)
	15 more	3 (42.9)	10.8 ± 3.8	
Fund support	yes	2 (28.6)	13.3 ± 1.1	3.5 (0.554)
	no	5 (71.4)	11.8 ± 3.1	
Research period	1–6 years	3 (42.9)	13.0 ± 1.3	5.0 (0.719)
	More than 6 years	4 (57.1)	11.6 ± 3.5	
Research field	Only exercise related	3 (42.9)	11.3 ± 4.2	6.0 (1.00)

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
