# Peer review of "A Methodological Quality Assessment of Meta-Analysis Studies in Dance Therapy Using AMSTAR and AMSTAR 2"

_healthcare, 2020, doi:10.3390/healthcare8040446_

Round 1
Reviewer 1 Report
This study aimed to evaluate the quality of meta-analysis studies in the field of dance therapy (DT) using the Assessment of Multiple Systematic Reviews (AMSTAR) and AMSTAR 2 assessment tools.
The authors concluded that:
1) Education on meta-analysis guidelines is required to improve the quality of DT-related meta-analysis studies,
2) Methodological caution is warranted, since different outcomes in evaluation scores for each tool may be obtained when using AMSTAR and AMSTAR2.
The manuscript is well written and present interesting findings. However, some modifications are required:
Figure 1 should be modified by a clearer figure.
If possible, I suggest that the authors in the revision verify if there is another paper that could be included and to update the date of the last search to the month of October.
I suggest, also, adding a figure in the procedure part explaining all steps of the evaluation.
Author Response
First of all, thank you very much for your screening. The contents of revision and supplementation according to the review are as follows
Response to reviewer 1’s comments
- Figure 1 should be modified with a clearer picture.
- To make the picture clearer, I increased the resolution and reproduced it.(line 160,254)
- If possible, check if there are other papers that the author of the revised may contain and suggest to update the last search date to October
- Our study was submitted on September 17, and it was judged that it was not appropriate for the period after the submission date to be described as the data collection period. Therefore, it was confirmed that July 30 was revised to August 30, and no additional papers were submitted until August 30.(line 147)
- It is advisable to add pictures to the procedural part that explains all the steps in the evaluation
- Figure 2, which can represent the process of the first half, was additionally inserted before the result.(line 225)
Look at the attached file

Reviewer 2 Report
The present study aims to evaluate the 16 quality of meta-analysis studies in the field of dance therapy (DT) using the Assessment of Multiple Systematic Reviews (AMSTAR) and AMSTAR 2 assessment tools. Meta-analysis studies on DT were collected from various databases. The methodology is robust. The whole article is well written. The authors found that Utilization of public education on meta-analysis guidelines is required to improve the quality of meta-analysis studies related to DT, while strict standards are required in the publication review process of meta-analysis studies. I would like to recommend its publication. The followings are some minor suggestions: First, I would like the authors to clarify/emphasize their aims in the last paragraph of the introduction, just like what has been described in the abstract. Second, please add the citation to line 59 for emphasizing the benefits of meta-analytic research: Utility of sonoelastography for the evaluation of rotator cuff tendon and pertinent disorders: a systematic review and meta-analysis. Eur Radiol. 2020.Author Response
First of all, thank you very much for your screening. The contents of revision and supplementation according to the review are as follows
Response to reviewer 2’s comments
- I hope the authors clarify/emphasize the purpose in the last paragraph of the introduction as explained in the abstract.
- The purpose of this study is clearly stated once again by adding the last paragraph of the introduction.(line 129)
- To highlight the benefits of meta-analysis studies, add a citation to line 59: The usefulness of ultrasound for the assessment of rotator cuff tendons and related disabilities: a systematic review and meta-analysis. Eur Radiol. 2020.
- Along with the 19th citation, citations on the benefits of meta-analysis were enriched by adding the citations from the research presented above. Also, for this purpose, the number of references to be entered has been modified overall.(line 61)
Look at the attached file

Reviewer 3 Report
Please see the attachment.

Author Response
First of all, thank you very much for your screening. The contents of revision and supplementation according to the review are as follows.
Response to reviewer 3’s comments
- I would like to understand a little more about the views of the authors in the specific field they study. I believe the author can write a little more on this topic in the discussion. For example, in the discussion you need to address:-What is the design for systematic review of your area?
- -When designing a statistical analysis or systematic review, what is the most appropriate protocol to follow in your area?
- -What are the future clinical research trends in your area?
- Thank you for your comment. However, this study is not suitable for the scope of this study to discuss the research trend of meta-analysis or design for review. We will take this well and solve it in subsequent studies.
- Overview studies for systematic review must be registered on a PROSPERO basis. Why was this registration not done? Authors can write their methodology in different databases such as PROSPERO, Open Science Framework (https://osf.io/), and Figshare (https://figshare.com/). Then insert your registration number into your methodology
- In this study, the range was selected for the database suggested by the methodology. In your opinion, I haven't been able to use databases like Prospero. I think this is a limitation of this study, but as far as we confirmed PROSPERO, it was confirmed that some studies related to dance therapy were not presented in PROSPERO. Therefore, it is judged that the database selected by us was able to collect more diverse data. Thanks for your comment. In future research, we will further expand the scope based on databases such as PROSPERO.
- If using percentages (including graphs), also enter absolute numbers
- Absolute numbers are indicated by presenting the frequency together with all values that appear using percentages.(line 254)
- In the case of `pubmed', it is actually spelled ``PubMed''.
- Modified the entire article, picture and table of PUBMED and Pubmed to be 'PubMed'.(line 138,160)
- Look at the attached file
